# Thickening and sickening the SYK model

## D. V. Khveshchenko

Department of Physics and Astronomy, University of North Carolina, Chapel Hill, NC 27599

⋆ khvesh@physics.unc.edu

## Abstract

We discuss higher dimensional generalizations of the $0 + 1$-dimensional Sachdev-Ye-Kitaev (SYK) model that has recently become the focus of intensive interdisciplinary studies by, both, the condensed matter and field-theoretical communities. Unlike the previous constructions where multiple SYK copies would be coupled to each other and/or hybridized with itinerant fermions via spatially short-ranged random hopping processes, we study algebraically varying long-range (spatially and/or temporally) correlated random couplings in the general $d + 1$ dimensions. Such pertinent topics as translationally-invariant strong-coupling solutions, emergent reparametrization symmetry, effective action for fluctuations, chaotic behavior, and diffusive transport (or a lack thereof) are all addressed. We find that the most appealing properties of the original SYK model that suggest the existence of its $1 + 1$-dimensional holographic gravity dual do not survive the aforementioned generalizations, thus lending no additional support to the hypothetical broad (including 'non-$AdS_{d+2}$/non-$CFT_{d+1}$') holographic correspondence.

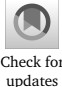
# 1 Original SYK model

Recently, there has been a lot of renewed interest in the fermionic variant [1,2] of the asymptotically soluble spin model with random, yet equally strong, all-to-all couplings which was originally proposed in the studies of quantum disordered spin-liquids [3–6] and which recently re-emerged as a much-needed test bed for the generalized holographic conjecture.

In its modern version the SYK model is described by the (Euclidean) action of a single $N$-colored Majorana fermion $S = \int_\tau (\sum_\alpha^N \chi^\alpha \frac{\partial}{\partial \tau} \chi^\alpha - H)$ governed by the Hamiltonian

$$H = i^{q/2} \sum_{\alpha_1 \ldots \alpha_q}^N J^{\alpha_1 \ldots \alpha_q} \chi^{\alpha_1} \cdots \chi^{\alpha_q}, \tag{1}$$

where the random $q$-fermion couplings (in the original Refs. [1, 2] one has $q = 4$) are completely antisymmetic with regard to the permutations of the 'color' indices $\alpha_i$. All the non-vanishing components with $\alpha_i \neq \alpha_j$ have zero mean and Gaussian variance

$$< J^{\alpha_1 \ldots \alpha_q} J^{\beta_1 \ldots \beta_q} >= \frac{J^2 (q-1)!}{N^{q-1}} \prod_a^q \delta^{\alpha_i \beta_i}. \tag{2}$$

By now this model has been extensively studied, although the exact nature of its conjectured gravity dual still remains somewhat elusive [7–17]. At first sight, on the bulk side of the purported correspondence it would suffice to have an asymptotically $AdS_2$ geometry in order to match such salient properties of the boundary SYK model as an asymptotic $0 + 1$-dimensional reparametrization invariance, power-law decay of the two-point correlation function, maximally chaotic behavior of the out-of-time-order (OTO) four-point one, etc. However, such a geometry alone does not uniquely identify the dual bulk theory as it would be common to all the $AdS_{d+1}$ gravitational backgrounds with a near-extremal horizon. Besides, the theories of gravity producing such geometry among their classical solutions are known to be plagued with such problems as UV-IR mixing and prone to a catastrophically strong backreaction.

In fact, despite an agreement between some of the SYK model's key thermodynamic as well as (in the generalized versions, see below) transport properties and the predictions made by using such a popular workhorse of the phenomenological ('bottom up') holography as the Einstein-dilaton(-Maxwell, if deformed away from the particle-hole symmetric point) gravity [18, 19], its prospective bulk dual is likely to lie outside the realm of the previously explored holographic scenarios.

Specifically, such a bulk dual is expected to have an infinite tower of scalar fields with nearly equidistant (yet, finite) masses. By contrast, in the regime of interest (i.e., for large values of the number of $N$ fermion colors and the properly defined t'Hooft coupling constant), the already established examples of holographic correspondence feature either only a finite (as in the best known and much studied duality between the maximally supersymmetric $SU(N)$ Yang-Mills gauge field theory and type-$IIB$ superstrings in $AdS_5 \otimes S^5$) or infinite (as in the $O(N)$ vector model/higher spin holography) number of massless bulk states. Therefore, in the would-be string-theoretical dual of the SYK model the fundamental string length and $AdS$ radius should be comparable, which property distinguishes it from those aforementioned $AdS/CFT$ examples that have long been providing inspiration for the various 'bottom up' holographic applications, including the ones claimed to be relevant to the various condensed matter systems [20–27].

To that end, such alternate descriptions as the Jackiw-Teitelboim gravity, $AdS_3$, and a certain $2 + 1$-dimensional Kaluza-Klein compactification have all been proposed as purported candidates for the bulk dual [28–36].

## 2 Generalized SYK-like models

While the original SYK model describing a single $O(N)$-symmetrical 'quantum impurity' in $0+1$ dimensions lacks any spatial dynamics, some of its early higher-dimensional generalizations allowed for a comparison with the available (e.g., transport-related) holographic predictions. In Refs. [13,37–54] this was achieved by introducing a $d$-dimensional array of the $L$ independent SYK models and hybridizing them, either by virtue of their direct (random) coupling or via some additional degrees of freedom defined on the thus-constructed spatial lattice.

Such 'SYK-chain' models were shown to exhibit the full-fledged diffusive modes corresponding to energy (as well as current, when the system is taken off the particle-hole symmetric point) transfer with the dispersion relation $\omega = i\mathcal{D}_\epsilon \mathbf{k}^2$, even in the absence of any propagating single-particle excitations. The latter are assumed to be described by the propagator

$$< \chi_i^\alpha(\tau)\chi_j^\beta(0) >= \delta^{\alpha\beta}G_{ij}(\tau), \tag{3}$$

which still remains ultra-local in space, $G_{ij}(\tau) \sim \delta_{ij}$.

Alternatively, in some of Refs. [37–54] the constituent fermions $\psi_i^\alpha$ ($\alpha = 1,\ldots,N; i = 1,\ldots,L$) were endowed with a $1d$ dispersion and mixed together with the use of a 'filter function' $f_{ij}$, the random SYK couplings then being imposed on their superpositions $\chi_i^\alpha = \sum_j f_{ij}\psi_j^\alpha$.

Yet another form of hybridization employed in Refs. [37–54] involves a single copy of the SYK 'impurity' coupled to a $1d$ chain of ordinary fermions and pairwise random couplings between the chain and the SYK fermions. Also, a supersymmetric generalization of the SYK model whose spectrum contains two-fermion bound states, alongside the fundamental fermions, was proposed in Ref. [55].

The before mentioned and similar 'semi-local' constructions allow one to study inter-site transport and compute the various kinetic coefficients, thereby putting to the test some of the general holographic predictions regarding the possibility of universal bounds for the kinetic coefficients and some simple relations between the transport and thermodynamic characteristics of the systems amenable to the holographic analysis [20–27].

Furthermore, the models of Refs. [37–54] reveal a number of interesting phase transitions from the manifestly non-Fermi liquid-like SYK state to the conventional Fermi liquid one or from a diffusive (ergodic) metallic to a (many-body) localized Mott insulator-type of state, as a relative number of the dispersionless Majoranas and ordinary ('conduction') fermions - and/or their coupling parameters - vary.

It is, however, unclear as to how (or even if) any of such behaviors could be captured in terms of a putative dual bulk picture and whether the corresponding changes in the bulk gravity would correspond to a 'capped' geometry terminating at a finite holographic radius, onset of non-locality, emergence of an 'incoherent' black hole, growth of higher-spin 'hair', or else. Besides, any viable holographic description of these models would be further hampered by such non-universal auxiliary elements as the aforementioned 'filter functions'.

Therefore, it is important to see if any of the holographic aspects of the original SYK model can survive a genuine, translationally-invariant, generalization to the higher dimensions without the customary assumption of ultra-locality of the two-point functions.

To that end, in the present communication we study a broad class of multi-dimensional (including asymptotically Lorentz-/Euclidean-invariant) SYK-like models with non-trivial spatial dispersion of the random couplings. Conceivably, such a spatial 'thickening' of the original single-site SYK model would allow one to explore some novel holographic scenarios, potentially lying outside of the conventional $AdS/CFT$ realm. Should this expectation indeed materialize, it could finally lend some support to the multitude of intriguing speculations (especially,

of the generalized 'non-$AdS$/non-$CFT$' kind) which have already become a popular lore in the 'bottom-up' holography over the past decade [20–27], albeit without solid justification.

We find, however, that such key properties of the original SYK model as emergent reparametrization invariance, maximally chaotic (yet, integrable) behavior, and existence of a bulk gravity dual do not survive the aforementioned generalizations under which the SYK model 'sickens', thus loosing much of its appeal as a potentially representative example of some more general holographic duality.

## 3 (Non-)local disorder correlations

Specifically, we consider the $D = d + 1$-dimensional action

$$S = \int_\tau \left( \sum_i^L \sum_\alpha^N \chi_i^\alpha \partial_\tau \chi_i^\alpha - i^{q/2} \sum_{i_a,\alpha_a} J_{i_1 \ldots i_q}^{\alpha_1 \ldots \alpha_q} \chi_{i_1}^{\alpha_1} \cdots \chi_{i_q}^{\alpha_q} \right), \tag{4}$$

where the Greek indexes stand for the $N$-valued colors, while the Latin ones run over the $L$ sites of a $d$-dimensional cubic lattice. The random $q$-fermion coupling amplitudes are now completely antisymmetric under the simultaneous permutations $\alpha_a \leftrightarrow \alpha_b$ and $i_a \leftrightarrow i_b$, with the Gaussian variance (here $\tau_{12} = \tau_1 - \tau_2$) for the non-vanishing components

$$< J_{i_1 \ldots i_q}^{\alpha_1 \ldots \alpha_q}(\tau_1) J_{j_1, \ldots j_q}^{\beta_1 \ldots \beta_q}(\tau_2) > = \frac{F_{i_1 \ldots i_q j_1, \ldots j_q}(\tau_{12})(q-1)!}{N^{q-1}} \prod_a^q \delta^{\alpha_a \beta_a}. \tag{5}$$

A uniform and spacetime-independent correlation function

$$F_{i_1 \ldots i_q j_1, \ldots j_q}(\tau_{12}) = J^2 \prod_a^q \delta_{i_a j_a} \tag{6}$$

corresponds to the original (single-site) SYK model of the total of $NL$ fermions where the correlations are equally strong between all the different sites and colors alike, while by choosing

$$F_{i_1 \ldots i_q j_1, \ldots j_q}(\tau_{12}) = J^2 \prod_a^q \delta_{i_a j_a} \prod_a^{q-1} \delta_{i_a i_q}, \tag{7}$$

one instead obtains $L$ decoupled copies of the $N$-fermion model.

For the sake of the following discussion it is worth commenting on the proper normalization of the variance in Eqs.(6,7) and alike. In the original SYK model defined on a dimensionless cluster of the total of $NL$ sites (where both $N$ and $L$ are treated as arbitrary integers) the right hand side of Eq.(6) would have to be multiplied with $1/L^{q-1}$ in order to have a well-defined 'large $NL$' limit.

However, in a setup that can be potentially relevant to the condensed matter applications the number of lattice sites $L$ would be macroscopic, whereas the typical number of 'orbitals' $N$ would be of order one. And even though N can still be formally considered large for the purpose of utilizing the soluble large-$N$ limit, treating $L$ in the same manner would amount to studying a mesoscopic 'quantum dot' where - as opposed to a macroscopically extended periodic $d$-dimensional spatial lattice - translational invariance is not an all-important factor and transport of charge, energy, and/or momentum (or a lack thereof) can not even be defined without introducing proper boundary conditions mimicking the external 'leads'. By contrast, in a macroscopic lattice system the strength of the random correlations would have to be chosen independent of the system's size $L$, thus allowing for a smooth behavior at $L \to \infty$ which is free of the finite-size effects.

In what follows, we focus on the 'interaction-like' correlations

$$F_{i_1...i_q j_1,...j_q}(\tau_{12}) = J^2_{i_q j_q}(\tau_{12}) \prod_a^{q-1} \delta_{i_a i_q} \delta_{j_a j_q},$$ (8)

which depend on the separation in space and/or time between the common locations of simultaneously disappearing and emerging $q$-fermion complexes. The non-vanishing components $J^{\alpha_1...\alpha_q}_{i...i}(\tau)$ still remain antisymmetric in the color space, though.

At first sight, the correlation (5) may seem to be quite different from the one that randomly couples and entangles $q/2$ fermions between an arbitrary pair of sites,

$$F_{i_1...i_q j_1,...j_q}(\tau_{12}) = J^2_{i_{q/2} i_q}(\tau_{12}) \prod_a^{q} \delta_{i_a j_a} \prod_a^{q/2-1} \delta_{i_a i_{q/2}} \prod_a^{q/2-1} \delta_{i_{q/2+a}, i_q}.$$ (9)

The latter ('entangling') type of correlation was previously utilized in the short-range-correlated one-dimensional 'SYK-chain' model of Refs. [13,42] where the function $J^2_{i_{q/2} i_q}(\tau_{12})$ takes non-zero values on the same $(J^2_{x,x}(\tau_{12}) = J^2_0)$ and adjacent $(J^2_{x,x\pm1}(\tau_{12}) = J^2_1)$ sites only.

However, it is easy to see that with the same choice of the function $J^2_{ij}(\tau_{12})$ the 'Fock-type' (or 'exchange') terms in the saddle-point Dyson-Schwinger equations (see Eq.(11) below) pertaining to the above two cases appear to be exactly identical, the only difference stemming from the additional 'Hartree-type' (or 'tadpole') terms which are present in the latter - but not the former - case.

Notably, in the original SYK model these two types of terms are indistinguishable due to the customarily assumed ultra-locality of the solutions (see Eq.(19) below), whereas any non-trivial dependence of $J^2_{ij}(\tau_{12})$ on the spatial distance $|i-j|$ allows one to discriminate between them.

Given the central role played by the saddle-point equation and its solutions in the analyses of Refs. [1–17] one then concludes that it is indeed the correlation law (8) that faithfully preserves the essential properties of the original SYK model and facilitates a comparative study of its generic multi-dimensional generalizations.

Moreover, in view of the special emphasis on the so-called 'hyperscaling violation' (HV) bulk geometries utilized in the great many holographic calculations [25–27], it would be particularly interesting to consider the functions $J^2_{ij}(\tau_{12})$ that decay algebraically in time and/or space, thereby prompting the use of manifestly non-local solutions. Conceivably, such power-law correlations might be a rather natural choice for constructing a SYK lattice whose putative bulk dual is characterized by one of the HV geometries with $z < \infty$.

## 4 Saddle-point equations

Upon averaging over a dynamical and/or spatially dispersive disorder under the usual assumption of replica-diagonal dynamics the theory (4) can be formulated as a functional integral over the two bi-local fields

$$Z = \int DG_{ij}(\tau) D\Sigma_{ij}(\tau) Pf(\partial_{\delta_{ij}\tau} - \Sigma_{ij}(\tau))$$

$$\exp\left(\frac{N}{2}\sum_{ij}\int_{\tau_1,\tau_2}\left(\frac{1}{q}J^2_{ij}(\tau_{12})G^q_{ij}(\tau_{12}) - G_{ij}(\tau_{12})\Sigma_{ij}(\tau_{12})\right)\right),$$ (10)

where $Pf$ stands for the Pfaffian determinant which results from integrating out the Majoranas.

The saddle points in the theory (10) are then described by the Schwinger-Dyson equation

$$\sum_k \int_{\tau_3} (\delta_{ik}\partial_{\tau_1}\delta(\tau_{13}) - \Sigma_{ik}(\tau_{13}))G_{kj}(\tau_{32}) = \delta_{ij}\delta(\tau_{12}), \tag{11}$$

where the self-energy is given, to the leading order in $1/N$, by the sum of the 'watermelon' diagrams

$$\Sigma_{ij}(\tau_{12}) = J_{ij}^2(\tau_{12})G_{ij}^{q-1}(\tau_{12}). \tag{12}$$

In the asymptotic conformal (IR) limit, the time derivative in (4) is outpowered by the self-energy and the discrete sum over the lattice sites can be replaced with the integral over the spatial coordinate **x**, thereby giving rise to the integral equation

$$\int_{\tau_3,\mathbf{x}_3} J_{\mathbf{x}_{13}}^2(\tau_{13})G_{\mathbf{x}_{13}}^{q-1}(\tau_{13})G_{\mathbf{x}_{32}}(\tau_{32}) = \delta(\tau_{12})\delta(\mathbf{x}_{12}). \tag{13}$$

The bosonic counterpart of these self-consistent equations have long been known [56]. It also emerges in the various disorder-free tensor models with the asymptotic SYK-like behavior [57–59].

In the case of the correlation function $J_{\mathbf{x}}^2(\tau) = const$ the strong-coupling equation (13) manifests an emergent reparametrization invariance under arbitrary diffeomorphisms $x^\mu \to f^\mu(x)$ of a flat $D$-dimensional spacetime ($x^\mu = (\tau,\mathbf{x})$), provided that the two-point function transforms as follows

$$G(x_1,x_2) \to |g(x_1)g(x_2)|^{1/2q}G(f(x_1),f(x_2)), \tag{14}$$

where $g = |det\,\partial f^\mu/\partial x^\nu|^2$ is the determinant of the metric $g_{\mu\nu}$ in the curvilinear coordinates $f^\mu$.

The $D = 1$-dimensional version of (14) known as $Diff(R^1)$ and its finite-temperature counterpart $Diff(S^1)$ were instrumental for solving the original SYK model. They also hint at the intrinsically geometric aspects of the underlying many-body dynamics which can be manifested via holography [1,2].

Choosing a concrete mean-field solution, however, breaks the symmetry (14) spontaneously, while the kinetic (time derivative) term in the action (4) violates it explicitly. Moreover, any nontrivial space- and/or time-dependence of the disorder correlator (5) does it as well. Unlike in the $D = 1$-dimensional case where the residual symmetry of the mean-field solution is $SL(2,R)$, though, the higher-dimensional symmetry-breaking patterns can be far richer.

It is also worthwhile contrasting (12) to the saddle-point equation obtained in the case of the 'entangling' correlation (9). The corresponding self-energy now includes, both, the aforementioned Hartree and Fock terms which are given, respectively, by the expressions (here we choose $q = 4$ for simplicity)

$$\Sigma_{ij}^{(H)}(\tau) = \delta_{ij}\sum_k J_{ik}^2(\tau)(G_{kk}^2(\tau)G_{ii}(\tau) + G_{ik}^2(\tau)G_{kk}(\tau)), \tag{15}$$

and

$$\Sigma_{ij}^{(F)}(\tau) = J_{ij}^2(\tau)(G_{ij}^3(\tau) + G_{ij}(\tau)G_{ii}(\tau)G_{jj}(\tau)). \tag{16}$$

As it has been already pointed out the two contributions become indistinguishable if the correlator is spatially local, $J_{ij}^2(\tau_{12}) \sim \delta_{ij}$.

## 5   Scaling analysis

A general applicability of the strong-coupling approach can be ascertained by invoking the standard scaling arguments. Under a scaling of the temporal, $\omega \to s\omega$, and spatial, $k \to s^{1/z}k$, dimensions, where $z$ is the dynamical critical index, one finds that in the weak-coupling regime the requirement for the kinetic term in (4) to be marginal endows the fermions with the engineering (UV) dimension $[\chi]_{UV} = d/2z$. As a result, the interaction term acquires the overall dimension $2q[\chi]_{UV} + 2[J] - 2 - 2d/z$, so that the condition guaranteeing that this term is $UV$-relevant and dominates over the kinetic one reads

$$\frac{d}{z}(q-2) + 2[J] - 2 < 0. \tag{17}$$

In the opposite, strong-coupling, limit the fermion dimension is generally expected to take a different (greater) value which would then be determined by the condition that, instead, the interaction term is marginal (hence, scale invariant), with the dimension $[\chi]_{IR} = (d/z + 1 - [J])/q$. Importantly, the requirement $[\chi]_{IR} > d/2z$ under which the interaction term continues to dominate over the (now, $IR$-irrelevant) kinetic one yields the very same criterion (17).

However, in the IR regime the dynamical index $z$ may take a different value as well, thus modifying this condition. In particular, the ultra-local SYK-like behavior stemming from the lack of a bare kinetic energy as a function of the spatial derivatives suggests $z_{UV} = \infty$, whereas an effective inter-site hopping developing in the IR regime promotes a finite $z_{IR} < \infty$. The condition of unitarity of the effective theory requires $z \geq 1$, though.

Notably, for $z_{UV} = \infty$ the condition (17) can be met for all $[J] < 1$. Same goes for the $q = 2$ model which, unlike in the general case of $q \geq 4$, is not chaotic.

However, for $z = 1$ and $q > 2$ the possibility of satisfying this criterion (even as equality) does not exist, except for $[J] = 0$, $D = 2$, and $q = 4$. Interestingly, in this case the fermions retain the same dimension $[\chi] = 1/2$ and the behavior of the system remains scale-invariant at all the time and distance scales.

The $D = 2$ Lorentz-invariant action proposed in Ref. [60] represents the chiral version of this model, the two independent $D = 1$ reparametrization symmetries acting on the separable light-cone coordinates $x^{\pm} = \tau \pm x$, respectively.

Moreover, similar to the related (albeit non-random and bosonic) $D$-dimensional model of Ref. [61], the kinetic term in (4) can be replaced with one of the higher order primary operators, such as $\chi \partial_{\tau}^n \chi$, in which case Eq.(17) changes to

$$\frac{d}{z}(q-2) + 2[J] - 2 - (n-1)q < 0, \tag{18}$$

thereby further extending the range of possibilities.

In that regard, the model of Ref. [60] presents an example of the $z = 1$ theory with the kinetic $n = 2$ term which emerges after the integration over an auxiliary bosonic field ('vierbein') which plays the role of a bound state of chiral fermions.

It is also feasible that a more detailed analysis of this marginal case could reveal logarithmic corrections resulting from a subtle violation of the underlying (near) conformal symmetry in the process of renormalization, as in the $D = 2$ random Thirring model of Ref. [61].

For all the other values of $z$, $q$ and $d$ the system can still attain its strong-coupling regime if correlations of the random amplitudes grow in space-time with a negative exponent, thus promoting the otherwise irrelevant higher-$q$ interaction term into a relevant one and satisfying the condition (17).

## 6  Mean-field solutions

In all the previous studies of the original SYK model and its generalizations [], it has been customary to resort to the ultra-local and spatially uniform saddle point (mean-field) solution

$$G_{ij}(\tau) \sim \frac{sgn\,\tau}{|\tau|^{2/q}} \delta_{ij}, \tag{19}$$

which breaks the symmetry (10) down to the subgroup $\prod_i \otimes SL(2,R)$ and accounts for the 'Hartree'-type spatial contractions (15). However, while being perfectly suitable in the case of a short-ranged disorder correlator, including $J_{ij}^2(\tau) \sim \delta_{ij}$, this solution may no longer be applicable once the correlations become sufficiently long-ranged.

Indeed, by plugging the solution (19) into Eq.(15) one finds that the sum $\sum_k J_{ik}^2(\tau)$ over the $d$-dimensional lattice is potentially IR-divergent. In particular, an algebraically decaying disorder correlation function (hereafter we use the lattice distance scale $a$ and choose the pertinent velocity to be equal unity)

$$J_{ij}^2(\tau) = J^2 (\frac{a}{\tau})^{2\alpha} (\frac{1}{|i-j|})^{2\beta} \tag{20}$$

would have given rise to a divergent spatial sum for $\beta \leq d/2$. On the other hand, the spurious UV divergence in the lattice sum for $\beta > d/2$ would, in fact, be absent as long as the amplitude $J_{ii}^2(\tau)$ remains finite.

This observation suggests that, at least, for $\beta \leq d/2$ the ultra-local solution (19) may become unstable and need to be replaced by a non-local ansatz $G_{ij}(\tau)$ that decays with the distance $\mathbf{x}_{ij} = a|i-j|$ while approaching a finite value at $i = j$, thus providing a regularization in the continuum limit $a \to 0$.

Having demonstrated the potentially problematic behavior - which would have been encountered with the use of the correlator (20), had any of the previous generalizations of the SYK model attempted to accommodate the long-range correlations with $\beta \leq d/2$ - we now return to our model where the self-energy (12) is free of the Hartree-type terms and, therefore, avoids the above complications altogether.

First, we consider the case of a product correlation function $J_{ij}^2(\tau) = F(\tau)H(\mathbf{x}_{ij})$ in which case the non-linear integral equation (13) admits factorized solutions $G_{ij}(\tau) = G(\tau)h(\mathbf{x}_{ij})$, the individual time- and space-dependent factors satisfying, correspondingly, the one-

$$\int_{\tau_3} F(\tau_{13})G^{q-1}(\tau_{13})G(\tau_{32}) = \delta(\tau_{12}), \tag{21}$$

and $d$-dimensional equations

$$\int_{\mathbf{x}_3} H(\mathbf{x}_{13})h^{q-1}(\mathbf{x}_{13})h(\mathbf{x}_{32}) = \delta(\mathbf{x}_{12}). \tag{22}$$

Combining an antisymmetric solution of Eq.(21) with a symmetric one of Eq.(22) we construct the overall antisymmetric space-time propagator.

Specifically, by using the correlator (20) one obtains the solution

$$G(\tau, \mathbf{x}) \sim \frac{sgn\,\tau}{\tau^{2\Delta_\tau}} \frac{1}{|\mathbf{x}|^{2\Delta_x}}, \tag{23}$$

where the exponents

$$\Delta_\tau = (1-\alpha)/q, \qquad \Delta_x = (d-\beta)/q, \tag{24}$$

can be readily read off by plugging the trial algebraic ansatz (23) into (13) and matching the power-laws on both sides. Notably, the symmetric solution of the $d$-dimensional Eq.(22) would also be relevant for the higher-dimensional bosonic counterpart of the model (4) discussed in Ref. [61].

The emergence of a non-local ('off-diagonal' in real space) component of the single-particle propagator can be viewed as an example of intrinsically non-perturbative spontaneous symmetry breaking. Naively, it would not have been generated in any finite order of perturbation theory built out of the purely local Eq.(19), had it not been for the aforementioned instability. As another - better known and well established - example of a single-particle 'condensate' we mention a finite density of states of the zero-density Dirac fermions in the presence of potential disorder which, while strictly vanishing in perturbation theory, signals spontaneous breaking of chiral symmetry above a finite coupling threshold in $d > 2$ and without any in $d \leq 2$.

The Fourier transform of Eq.(23) factorizes, too, thus yielding a product of the two-point functions in the frequency and momentum domains which conforms to the generally anticipated expression

$$G(\omega, \mathbf{k}) \sim \omega^{-\eta} \rho(\omega/k^z), \tag{25}$$

with $\eta = 1 - 2\Delta_\tau + (d - 2\Delta_x)/z$. However, the dynamical critical exponent $z$ remains arbitrary and can not be unambiguously determined without putting the subdominant kinetic term back into Eq.(13). With this term added, though, the position of the pole of $G(\omega, \mathbf{k})$ scales with a finite momentum as a power

$$z = \frac{d(q-2) + 2\beta}{2(1-\alpha)}, \tag{26}$$

that should be viewed as the limiting value of $z$ for which the criterion (17) can still be satisfied, albeit only marginally. In this case neither the anomalous self-energy, nor the bare kinetic term dominate, while keeping both can alter the numerical prefactor in (25) by a coefficient of order one. Concomitantly, the power-law prefactor is then governed by $\eta = 1$ as it should be in the absence of an alternate energy scale.

In particular, for $q = 2$ one then obtains $z = \beta/(1-\alpha)$. The form of the universal function $\rho(\omega/k^z)$ in (25) can then be readily found by solving the (now becoming quadratic) equation (13), thus resulting in

$$\rho(u) = u - \sqrt{u^2 - 1}. \tag{27}$$

In the exactly solvable case of static infinitely-range-correlated disorder ($\alpha = \beta = 0$) the imaginary part of this function exhibits the well known 'semi-circular law'.

For generic values of $q$ and $d$ and in the case of spatially-independent disorder with $\beta = 0$ the solution (23) behaves as

$$G(\tau, \mathbf{x}) \sim sgn\tau \, \tau^{-2(1-\alpha)/q} |\mathbf{x}|^{-2d/q}, \tag{28}$$

while in the complimentary case of a time-independent correlator with $\alpha = 0$ one gets

$$G(\tau, \mathbf{x}) \sim sgn\tau \, \tau^{-2/q} |\mathbf{x}|^{-2(d-\beta)/q}. \tag{29}$$

To contrast the above solutions against the ultra-local Eq.(19) one can also evaluate the corresponding energies. To the leading mean-field approximation the energy is given by the expression in the exponent in Eq.(10) with the self-energy taken from (12)

$$E = \frac{q-1}{2q} \int_{\tau, \mathbf{x}_1, \mathbf{x}_2} J^2_{\mathbf{x}_{12}}(\tau) G^q_{\mathbf{x}_{12}}(\tau). \tag{30}$$

Adjusting the proper normalization factors in the propagators given by Eqs. (19) and (23) (whose ratio approaches $d^{1/q}$ at large $d$) one finds that the mean-field energies of both solutions are exactly equal.

Notably, even if the limits $\alpha, \beta \to 0$ were taken, thus making the disorder correlations time- and space-independent, the action (4) defined on an arbitrary $L$-site lattice would still differ from the original SYK model of the total of $NL$ species. Indeed, each of its terms can only entangle fermions between one pair of sites, contrary to the completely unrestricted (all colors, all sites) couplings (6) (besides, the latter couplings would have to be renormalized by the factor $1/L^{q-1}$ before taking the $L \to \infty$ limit and making such a comparison). Therefore, it should not come as a surprise that the non-local solution (23) remains markedly different from (19) even in the above limit.

Another expressly non-local solution of Eq.(13) can be found in the case of a Lorentz-(Euclidean-)invariant kernel

$$J_{ij}^2(\tau) = J^2 \left( \frac{a^2}{\tau^2 \mp a^2 |i-j|^2} \right)^{\gamma}. \tag{31}$$

While factorization is no longer possible, the solution inherits the same structure as the kernel

$$G(\tau, \mathbf{x}) \sim \frac{sgn x^-}{(x^+ x^-)^{\Delta}}, \tag{32}$$

where $x^{\pm} = \tau \pm |\mathbf{x}|$, thus being characterized by $z = 1$. In particular, the propagator (32) emerges in the Lorentz-invariant model of Ref. [60] with $n = 2$, $d = 1$, and $\gamma = 0$.

Plugging the algebraic ansatz (32) into Eq.(13) one now obtains

$$\Delta = (D - \gamma)/q, \tag{33}$$

while the corresponding Fourier transform reads $G(\omega, \mathbf{k}) \sim (\mathbf{k}^2 \mp \omega^2)^{\Delta - D/2}$.

Importantly, Eqs.(23) and (32) remain different from one another even if the limits $\gamma \to 0$ and $\alpha \to 0, \beta \to 0$ are taken, respectively. In that regard, they should be viewed as two (out of, possibly, many) different patterns of spontaneously breaking the local $Z_2$ symmetry $\psi_i^{\alpha} \to -\psi_i^{\alpha}$ of the action (4).

Besides, the solutions (23) and (32) can not be even formally valid for arbitrary values of the parameters. Specifically, the applicability of Eq.(32) is restricted by the condition (17) with $z = 1$, thus requiring $\gamma < 0$ if $d > 0$ and $q > 2$, at least, for $n = 1$.

Taken at its face value, this observation implies that the random amplitudes' correlator must increase as a function of the interval. Such a behavior would generally be incompatible with unitarity and, therefore, the corresponding strongly coupled IR theory may not be well defined (this observation is corroborated by the conclusions reached in Ref. [63]).

## 7 Fluctuations about mean-field solutions

The fluctuations about any chosen mean-field solution $G_0$ can be studied by introducing the expansions $G = G_0 + g|G_0|^{(2-q)/2}$ and $\Sigma = \Sigma_0 + \sigma|G_0|^{(q-2)/2}$ and deriving the '$\sigma$-model' action (here the integrations are performed over the continuous $D$-dimensional coordinates $x = (\tau, \mathbf{x})$)

$$\delta S(g, \sigma) = N \int_{x_1, x_2} \left( g_{12}\sigma_{12} - \frac{q-1}{2} J_{12}^2 g_{12}^2 \right) - \int_{x_1, x_2, x_3, x_4} \frac{\sigma_{12}\hat{K}_{12,34}\sigma_{34}}{2(q-1)}, \tag{34}$$

from which, upon integrating out $\sigma$, one arrives at the quadratic form

$$\delta S(g) = \frac{N(q-1)}{2} \int_{x_1, x_2, x_3, x_4} g_{12}(\hat{K}_{12,34}^{-1} - \hat{1}_{13}\hat{1}_{24} J_{12}^2) g_{34}, \tag{35}$$

with the symmetrized 'conformal' kernel

$$\hat{K}_{12,34} = (q-1)|G_{12}G_{34}|^{\frac{q-2}{2}}G_{13}G_{24}. \tag{36}$$

In the Lorentz-invariant case with the correlator (31), solutions to the eigen-function equation

$$\int_{x_3,x_4} \hat{K}_{12,34}J_{34}^2\Psi_{34}(h,s|\omega,\mathbf{k}) = \lambda_{h,s}(\omega,\mathbf{k})\Psi_{12}(h,s|\omega,\mathbf{k}) \tag{37}$$

can be sought in the form

$$\Psi_{12}(h,s|\omega,\mathbf{k}) \sim (x_{12}^+)^{h_R/2-\Delta}(x_{12}^-)^{h_L/2-\Delta}e^{ik_\mu(x_1^\mu+x_2^\mu)/2}, \tag{38}$$

where $k_\mu = (\omega,\mathbf{k})$ is the $D$-dimensional covariant momentum, while $h_{R,L}$ are the right/left dimensions of an operator with the total dimension $h = (h_R+h_L)/2$ and spin $s = (h_R-h_L)/2$ from the product expansion of a pair of fundamental fermions.

In the original SYK model, the kernel (36) is related to the Casimir operator of the conformal $SL(2,R)$ algebra which yields the masses $m^2 = h(h-1)$ of the tower of bulk fields dual to the primary (spinless) operators $\chi\partial^{2n+1}\chi$ with the dimensions approaching $h_n = 2\Delta+2n+1$ [7–10].

Correspondingly, the spectrum of Eq.(37) consists of a continuum of 'scattering' ($h = 1/2+is$) and a discrete set of 'bound' ($h = 2,4,6,\dots$) states. In particular, the emergent reparametrization invariance gives rise to the zero mode of dimension $h = 2$ which is identified with the fluctuations of stress-energy tensor. Its presence is considered to be a strong argument in favor of the existences of a (local) gravitational description of the corresponding putative theory in the bulk [1, 2, 7–10].

In the case of the generalized $D$-dimensional Lorentz-invariant theory with the aforementioned disorder correlator (31) and for $k_\mu = 0$ Eq.(37) takes the form

$$\frac{\lambda_{h,s}}{x_{12}^{2\Delta-h}}\left(\frac{x_{12}^+}{x_{12}^-}\right)^s = \int_{x_3,x_4}\frac{sgn(x_{13}^-x_{24}^-)}{x_{13}^{2\Delta}x_{24}^{2\Delta}x_{34}^{2D-2\Delta-h}}\left(\frac{x_{34}^+}{x_{34}^-}\right)^s, \tag{39}$$

where $\lambda_{h,s} = \lambda_{h,s}(0,\mathbf{0})$, and the pertinent values of $h_{L,R}$ are given by the roots of the equation $\lambda_{h,s} = 1$. Calculating the integrals in (39) one obtains

$$(1-q)\frac{\Gamma(\frac{3D}{2}-\Delta)\Gamma(D-\Delta)\Gamma(\frac{h_L}{2}+\Delta)\Gamma(\frac{D}{2}+\Delta-\frac{h_R}{2})}{\Gamma(\frac{D}{2}+\Delta)\Gamma(\Delta)\Gamma(\frac{3D}{2}-\Delta-\frac{h_R}{2})\Gamma(D-\Delta+\frac{h_L}{2})} = 1. \tag{40}$$

Notably, for $s = 0$ the function $\lambda_{h,0}$ remains symmetric under $h \leftrightarrow D-h$ and becomes constant ($\lambda_{h,0} = -1$) for $q = 2$.

Eq.(40) can also be contrasted against its bosonic (symmetric) counterpart akin to that discussed in Ref. [64]. The minimal kinetic term of the bosonic analogue of (4) contains two time derivatives and, therefore, the criterion of a non-trivial IR behavior (17) where the choice of $n = 2$, $z = 1$, and $[J] = 0$ limits the acceptable values of the remaining parameters to $q = 4$ and $d < 4$.

The corresponding equation then reads

$$(1-q)\frac{\Gamma(D-\Delta)\Gamma(\frac{D}{2}-\Delta)\Gamma(-\frac{D}{2}+\Delta+\frac{h_L}{2})\Gamma(\Delta-\frac{h_R}{2})}{\Gamma(-\frac{D}{2}+\Delta)\Gamma(\Delta)\Gamma(D-\Delta-\frac{h_R}{2})\Gamma(\frac{D}{2}-\Delta+\frac{h_L}{2})} = 1. \tag{41}$$

As far as the scalar ($s = 0$) operators are concerned, for $D = 4-\epsilon$ Eq.(41) has the complex-valued roots $h_0 = 2\pm i(6\epsilon)^{1/2}+\dots$, $h_1 = 4+\epsilon+\dots$, and $h_n = 2(n+1)-\epsilon/2+\dots$ for

$n > 1$, thus signaling potential instabilities driven by the corresponding composite two-fermion operators [64].

More generally, for $D < 4$ the solution of (41) with the lowest real part is $h_0 = D/2 \pm iO(1)$. In particular, for $D = 1$ the first normalizable mode has $h_0 \approx 1/2 \pm 1.5i$ while for $D = 2$ it is $h_0 \approx 1 \pm 1.5i$. However, Ref. [64] finds that for $D > D_{cr} \approx 1.9$ there exists a critical value $q_{cr} \approx 4/(D - D_{cr})$ such that for $q > q_{cr}$ the solutions are all real and the theory does not develop any two-particle instabilities.

As far as the non-scalar operators of spin $s > 0$ are concerned, in the reparametrization-invariant case of $\gamma = 0$ (hence, $\Delta = D/q$) both Eqs.(40) and (41) feature solutions with $h = D$ and $s = 2$ which (not unexpectedly) correspond to the stress-energy tensor.

However, for any non-zero $\gamma$ (which may be required to meet the criterion (17)) the reparametrization symmetry would be broken and fluctuations of the stress-energy tensor would not necessarily remain a massless mode, hence the above solution would not survive.

Moreover, for the corresponding pattern of the reparametrization symmetry breaking there might not be any residual $SL(2,R)$ symmetry left, so the scaling dimensions solving Eqs.(40,41) may not be simply related to the eigenvalues of any invariant operator, as in the original SYK model [7–10].

Furthermore, the reparametrization mode does not necessarily dominate the operator expansion of the product of fermion fields (and, correspondingly, gravity may not be the main player in the bulk theory). Full implications of this observation for the existence of a (local) holographic bulk dual remain to be ascertained (to that end, certain indications in favor of a non-gravitational bulk theory were discussed in Ref. [65]).

Regardless of the possibility of a bulk interpretation, though, the explicitly broken reparametrization symmetry gives rise to a non-trivial effective action for the pseudo-Goldstone modes corresponding to the time- and/or space-dependent fluctuations of the collective fields $G$ and $\Sigma$. In the original SYK model, such action can be obtained by directly expanding the Pfaffian, the result being given by the Schwarzian derivative associated with the mapping $\tau \to f(\tau)$

$$\delta S(f) = \frac{N}{J} \int_\tau \{f, x\} = \frac{N}{J} \int_\tau \left( \frac{f'''}{f'} - \frac{3}{2} \left( \frac{f''}{f'} \right)^2 \right), \tag{42}$$

whose manifestly geometric appearance was recognized early on as a strong indication in favor of the underlying holographic connection to the bulk gravity.

Likewise, in the Lorentz-invariant $D = 2$ model of Ref. [46] with the $n = 2$ kinetic term one finds the universal action

$$\delta S(f^\mu) = N \sum_{\mu, \nu = \tau, \mathbf{x}} \int_x det\{f_\mu, x_\nu\}, \tag{43}$$

which, with the use of the light-cone coordinates $x^\pm$, amounts to the product of two Schwarzian factors for the left and right chiral fermions $\chi_{R,L}(x^\pm)$. Another salient feature of Eq.(43) is the absence of an external energy scale (cf. Eq.(42)).

However, in the general case of long-ranged space/time disorder correlations a disorder-related contribution to the effective action for the fluctuations may dominate over that generated by the (now, IR-irrelevant) kinetic terms, thereby producing a non-universal and non-geometrical expression instead of (43).

Following the procedure performed in the original SYK model [7–10] and computing the action functional (35) on the variation

$$g_{12}(\epsilon^\mu) = [\Delta(\partial_1^\mu \epsilon_1^\mu + \partial_2^\mu \epsilon_2^\mu) + \epsilon_1^\mu \partial_1^\mu + \epsilon_2^\mu \partial_2^\mu]G_{1,2} \tag{44}$$

as a function of the infinitesimal coordinate transformation $f^\mu(x) = x^\mu + \epsilon^\mu(x)$, one finds the leading quadratic term which, to first order in $\alpha$ and $\beta$ and at zero temperature, takes the

following schematic form

$$\delta S(\epsilon^\mu) = \frac{N}{2a^d} \int_{\omega,\mathbf{k}} \left( |\omega|\delta + \frac{\omega^2}{J} \right) \epsilon_\mu(\omega,\mathbf{k}) M_{\mu\nu} \epsilon_\nu(-\omega,-\mathbf{k}), \tag{45}$$

where the matrix elements $M_{\mu\nu}$ are symmetric quadratic forms of $\omega$ and/or $\mathbf{k}$, all the non-vanishing coefficients being of order one. Obtaining their accurate value, alongside the overall 'eigenvalue shift' $\delta = O(\alpha) + O(\beta)$, could be rather difficult, although the linear dependence of the latter upon $\alpha$ and $\beta$ (at least, if those are small) is rather straightforward as it stems from breaking the $D$-dimensional reparametrization invariance by a temporal (for $\alpha \neq 0$) and/or spatial (for $\beta \neq 0$) dependence of the disorder correlations. In contrast to the bare kinetic term, though, such symmetry breaking occurs at arbitrarily large $J$ and, therefore, the corresponding term lacks the extra $\omega/J$ factor, as compared to the case of the original SYK, thus becoming dominant in the IR regime.

At a finite temperature $T$ the square of the time derivative of $\epsilon_\tau$ gets replaced with $(|\omega|^2 - (2\pi T)^2)|\epsilon_\tau|^2$, thus softening the spectra of the two additional zero modes $\epsilon_\tau(\pm 2\pi T, \mathbf{k} = 0)$.

It is worth mentioning that the effective action (45) for the fluctuations about the non-local mean-field solution (23) differs from that derived by expanding about the ultra-local one (19). From the technical standpoint, in the latter case the only relevant soft mode is $f_\tau(\tau, \mathbf{x})$ where the spatial coordinate pertains to the center-of-mass of a pair of fermions, as opposed to the relative distances between them, as in Eq.(45).

This time around, by computing (35) on the fluctuation (44) one obtains

$$\delta S = \frac{N}{2a^d} \int_{\omega,\mathbf{k}} |\omega|^3 \left( \alpha + \xi_\mathbf{k} + \frac{|\omega|}{J} \right) |\epsilon_\tau|^2, \tag{46}$$

where the lattice sum

$$\xi_\mathbf{k} = \sum_{<i,j>} \frac{J_{ij}^2}{J^2} (1 - \cos \mathbf{k}\mathbf{x}_{ij}) \tag{47}$$

behaves as $\sim \mathbf{k}^2$ at small $\mathbf{k}$ only for $\beta > (d+2)/2$ (and, of course, for any faster-than-algebraic decaying spatial correlations), whereas for $d/2 < \beta < (d+2)/2$ it becomes $\sim \mathbf{k}^{2\beta-d}$. At still smaller values of $\beta$ the sum is IR-divergent, thus hinting at a problematic thermodynamic limit.

Therefore, apart from the constant term $O(\alpha)$ the 'eigenvalue shift' demonstrates a generally non-quadratic momentum dependence. For $d = 1$ this last point was also made in Ref. [51], although such a behavior was claimed to occur for $\beta < 1$, rather than $\beta < 3/2$, which conclusion resulted from erroneously substituting $Li_{2\beta}(\cos ka)$ for $ReLi_{2\beta}(e^{ika})$.

Although the above conclusions pertain to the specific family of disorder correlations described by Eqs.(8) and (20), this class of functions is sufficiently broad to suggest that even more general long-range correlations decaying slower than a certain critical power-law (evaluated above as $(d+2)/2$) could give rise to the behavior that is markedly different from that obtained in the generic short-range correlated 'SYK-lattice' models, including those with the 'same and nearest neighbor only' [13,42] as well as exponentially decaying random couplings.

For comparison, in the case of the Lorentz-invariant disorder correlator (31), the corresponding effective action takes the schematic form

$$\delta S = \frac{N}{2a^d} \int_{\omega,\mathbf{k}} \left( \gamma a^d (k_\lambda^2)^{D/2} + \frac{\omega^2}{J} \right) \epsilon_\mu(\omega,\mathbf{k}) M_{\mu\nu} \epsilon_\nu(-\omega,-\mathbf{k}). \tag{48}$$

The Lorentz invariance restricts the structure of the matrix elements to the two invariant combinations $M_{\mu\nu} = Ak_\mu k_\nu + B\delta_{\mu\nu} k_\lambda^2$ in terms of the covariant momentum $k_\mu = (\omega, \mathbf{k})$. Likewise,

the universal (first) term in the first factor in (48) ought to be manifestly Lorentz-invariant, hence a function of the sole invariant argument $k_\lambda^2$.

In accordance with the earlier discussion, though, the Lorentz-invariant term dominates over the non-invariant one stemming from the $1^{st}$ order time derivative for $D < 2$ only (or, else, at some intermediate energies), thus limiting the practical relevance of such regime.

## 8 Diffusive transport and chaotic properties

An effective action for the soft reparametrization modes allows one to study correlations of the stress-energy tensor $T_{\mu\nu} = \delta S/\delta \partial_\mu f_\nu$. Focusing on the component $T_{\tau\tau}$ which represents energy density, in the short-range-correlated SYK-chain model of Refs. [13, 42] one obtains

$$< T_{\tau\tau}(\omega, \mathbf{k}) T_{\tau\tau}(-\omega, -\mathbf{k}) > = \frac{|\omega|}{J} \frac{(2\pi T)^2 - |\omega|^2}{|\omega| + \mathcal{D}_\epsilon \mathbf{k}^2}, \tag{49}$$

which features (upon the Wick rotation to real frequencies, $\omega \to i\omega$) a diffusion pole at $\omega = i\mathcal{D}_\epsilon \mathbf{k}^2$ with $\mathcal{D}_\epsilon \sim J$. In Refs. [13, 39, 41, 42, 46] the expression (49) would be further forced into the conventional diffusion propagator by subtracting its value at $\mathbf{k} = 0$ and discarding the term $\sim \omega^3$ in the numerator.

Moreover, in the SYK-chain model of Refs. [13, 42] the corresponding diffusion coefficient was shown to saturate the previously conjectured bound [29] $\mathcal{D}_\epsilon > v_B^2/2\pi T$ which is considered to be characteristic of a fully thermalized and maximally chaotic behavior. Whether or not - and, if so, under what circumstances - this bound continues to hold for $d > 1$ remains to be seen.

In contrast, the long-range correlated disorder (20) gives rise to the gapped behavior

$$< T_{\tau\tau}(\omega, \mathbf{k}) T_{\tau\tau}(-\omega, -\mathbf{k}) > = \frac{|\omega|}{Ja^d} \frac{((2\pi T)^2 - |\omega|^2)}{(|\omega| + J\delta)}, \tag{50}$$

thus signaling a lack of the ordinary energy diffusion.

In turn, the Lorentz-invariant long-range-correlated disorder (31) yields

$$< T_{\tau\tau}(\omega, \mathbf{k}) T_{\tau\tau}(-\omega, -\mathbf{k}) > = \frac{|\omega|^2}{Ja^d} \frac{((2\pi T)^2 - |\omega|^2)}{(|\omega|^2 + J\gamma a^d (k_\mu^2)^{D/2})}, \tag{51}$$

which behavior is markedly different from the conventional diffusion as well.

In the SYK model, the energy diffusion was shown to coexist with a maximally chaotic behavior. The latter (or a lack thereof) can be detected by analyzing the four-point function

$$\mathcal{F}_{12,34}^{\alpha\beta\gamma\delta} = < \chi_i^\alpha(\tau_1) \chi_j^\beta(\tau_2) \chi_k^\gamma(\tau_3) \chi_l^\delta(\tau_4) > \tag{52}$$

which, in the large-$N$ limit, is given by the sum of the ladder diagrams

$$\mathcal{F}_{12,34} = \frac{1}{1-\hat{K}} |\mathcal{F}^0 > = \sum_\lambda \Psi_{12} \frac{1}{1-\lambda} < \Psi_{34} |\mathcal{F}^{(0)} >, \tag{53}$$

where $\mathcal{F}_{12,34}^{(0)} = G_{13} G_{24} - G_{14} G_{23}$.

At a finite temperature, the properly parsed Eq.(52) can be identified with the OTO amplitude

$$\mathcal{F}(\tau, \mathbf{x}) = < u\chi_\mathbf{x}^\alpha(\tau) u\chi_\mathbf{0}^\beta(0) u\chi_\mathbf{x}^\alpha(\tau) u\chi_\mathbf{0}^\beta(0) >, \tag{54}$$

where $u = e^{-H/4T}$, thereby providing an informative chaos marker.

Inverting the properly defined ('retarded', [7–10]) counterpart of the integral kernel (36) involves a finite-$T$ generalization of the spin $s = 0$ eigenfunctions (38)

$$\Psi_{12}(h_r|\mathbf{k}) \sim \frac{e^{i\mathbf{k}(\mathbf{x}_1+\mathbf{x}_2)/2 - \pi T h(\tau_1+\tau_2)}}{\cosh(\pi T \tau_{12})^{2\Delta_\tau - h_r}|\mathbf{x}_1 - \mathbf{x}_2|^{2\Delta_x - h_r}} \tag{55}$$

In the original SYK model ($D = 1$ and $\Delta_\tau = 1/q$) the corresponding eigenvalue equation

$$\frac{(q-1)\Delta_\tau}{(1-\Delta_\tau)} \frac{\Gamma(3-2\Delta_\tau)\Gamma(2\Delta_\tau - h_r)}{\Gamma(1+2\Delta_\tau)\Gamma(2-2\Delta_\tau - h_r)} = 1 \tag{56}$$

has the negative root $h_r = -1$, thus leading to the maximally chaotic (fastest scrambling) behavior, as manifested by the intermediate asymptotic of the correlation function (54)

$$\mathcal{F}(\tau, \mathbf{x}) \sim 1 - \frac{1}{N} e^{\lambda_L(\tau - |\mathbf{x}|/v_B)}, \tag{57}$$

which exhibits the saturation of the Lyapunov exponent $\lambda_L = -2\pi T h_r = 2\pi T$ and the 'butterfly' velocity of chaos spreading $v_B = (2\pi \mathcal{D}_\epsilon T)^{1/2}$ [7–10].

It should be noted, though, that had it not been for the first factor in (56) which reflects the $\Delta$-dependent normalization of the two-point function, this equation would have had the root $h_r = -1$ for all $\Delta_\tau$. However, for $D > 1$ and $\mathbf{k} = 0$, apart from this factor Eq.(56) also contains the spatial integral which coincides with the left hand side of the bosonic eigenvalue Eq.(41) where the substitutions $D \to d$ and $\Delta \to \Delta_x$ are to be made.

As a result, one finds that for the disorder correlators (20) and (31) the largest allowed negative eigenvalue $h_r = -1$ is no longer a solution. Therefore, the system's state still remains chaotic, albeit not maximally so (this observation agrees with the conclusions drawn in Refs. [38–40, 44, 52] for $D = 2$).

While the above conclusions hold in the large-$N$ limit it is worth mentioning the contrasting report of an apparent insulator-to-metal transition with the increasing range of correlations $l$ in the $N = 1$, $L \to \infty$ (in our notations) SYK-chain-like model with the step-wise disorder correlator $J_{ij}^2 = J^2 \theta(l - |i - j|)$ [66]. Similar to the earlier works on the related topic, such a transition was detected by a marked change in the level statistics from the Poissonian (indicative of a chaotic metallic state) to the Wigner-Dyson ensemble hinting at the formation of a (many-body?) localized insulator.

Thus, the accumulated evidence suggests that the maximum chaos (alongside the conjectured holographic correspondence), too, is not a universal feature of the generalized SYK-like models but, in fact, can be highly sensitive to such important details as the range of disorder correlations, their Lorentz (non-)invariance, spatial dimension, the ratio $N/L$, etc.

## 9 Conclusions

To summarize, our attempt of a non-ultra-local generalization of the SYK model to higher dimensions - with or without the (asymptotic) Lorentz symmetry present - shows that such a 'thickening' of the original $D = 1$-dimensional model also tends to 'sicken' it, thus making it less likely for the system in question to develop an emergent multidimensional reparametrization symmetry, alongside such key properties as a geometric nature of the effective action for fluctuations, diffusive transport, fully thermalized chaotic behavior, etc.

More specifically, attaining a strong-coupling regime in $D > 1$ dimensions requires the presence of long-ranged time- and/or spatially-dependent correlations of the random $q$-fermion

amplitudes which, in turn, appear to be incompatible with the invariance under general diffeomorphisms. Thus, although the original SYK model itself might indeed have a well-defined holographic dual, this particular form of the holographic correspondence does not appear to be readily extendable into higher dimensions in the general case of time/distance-dependent disorder correlations.

The fact that the unique properties of the original SYK model do not appear to survive such deformations suggests that it is unlikely to provide a suitable framework for the putative holographic analysis of those physical systems whose behavior is not ultra-local.

It has been argued, though, that the $AdS_2$-based bulk theory might still offer an adequate description of a sufficiently large class of the experimentally relevant systems, inclusive of the cuprates, heavy fermions, and other examples of the Kondo-type ($z = \infty$) physics [25, 26], as well as the various mesoscopic and cold atom setups proposed for experimental simulations of the SYK Hamiltonian [67–72]. The salient features of this universality class are likely to include such common behaviors as linear resistivity and saturated bounds for certain kinetic coefficients and their ratios.

Our findings suggest, however, that outside the above class of systems the SYK model may not immediately provide a still badly needed justification for the numerous 'bottom-up' (for their most part, 'non-$AdS$/non-$CFT$') holographic constructions which involve the bulk geometries of a far broader variety then the near-$AdS$ one - e.g., the hyperscaling violation metrics that are potentially capable of producing continuously varying (hence, seemingly non-universal) critical exponents [20–27] - and so the true status of such studies still remains hard to ascertain.

If proved to be valid and firmly justified in the context of realistic solvable models, the holographic phenomenology would indeed become an invaluable advanced phenomenological framework for discovering new and classifying the already known types of the NFL behavior. However, for the time being, the ultra-local SYK model and its short-ranged generalizations appear to remain in the league of their own, thus offering an important, yet somewhat limited, insight into the realm of higher-dimensional non-Fermi liquids.

**Note added:**     This report was first released as arXiv:1705.03956.

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
