# Peer review of "Thickening and sickening the SYK model"

_SciPost Physics, doi:SciPost Phys. 5, 012 (2018)_

## Round 3 · Referee Report · Anonymous (Referee 1) · 2018-3-22

Strengths

  1. In this paper the author considers different kinds of long-range correlated generalizations of the original SYK model, including different dimensions, dynamical critical index z and the range of the random interactions. A thorough power counting is given to search for possible set-ups for the existence of scale-invariant solutions.
  2. The author gives the effective action for the fluctuation near the saddle point solution and discusses the transport and chaotic behaviors of these systems, which shows interesting new behaviors.
  3. Along with the field theory calculation, relations and implications on the bulk side are discussed. The results suggest the SYK model only provides limited insights into AdS-CFT correspondence, which is reasonable since the (maximally chaotic) SYK set-up mainly works for $z=\infty$.

Weaknesses

  1. When the correlation function of the interaction strength is given by (6), the model is reduced to an SYK model with $NL$ fermions only if $J\propto1/L^{3/2}$. This also leads to the divergence when one combines (20) with (15), which can be canceled by choosing a correct scaling of $J$ in terms of $L$. As a result the instability discussed below (20) may not be physical. (By the way, even if we do not add a scaling of $L$ to $J$, I think the divergence can not be canceled by assuming the non-locality of Green's function.)
  2. If we take Eq. (10) and use a diagrammatic approach starting from a non-interacting Green's function and summing up all diagrams, the Green's function will always be super-local. The author does not provide any evidence for the emergence of the non-locality (The overall factor of the Green's function is not given and the author does not compare the energy of different solutions). Or maybe the simplest way is just modifying the UV kinetic energy by adding some spatial derivatives to tune different z. Will this modify the effective action?
  3. The Eq. (26) is derived by considering the kinetic term, but the Green's function is solved after neglecting the kinetic term. The logic here may need further explanation.
  4. Some results (for example Eq. (45) and (48)) are given without derivation, which makes the paper less easy to understand. Taking (45) as a example, does the result means: the coefficient of the $\delta$ term given by the inner product of wave function and the $1/J$ terms indicates the shift of eigenvalue of $K$ by the UV kinetic term is proportional to $|\omega|$ (as the original SYK model)? Is there a simple scaling argument for the form of them?
  5. There are some typos in the draft. For example: below Eq. (20) there is an empty bracket "()".

Report

Recently, the SYK model, which is a 0+1-D solvable model for chaotic non-Fermi Liquid and may be related to holographic duality, attracts much attention. Later, the model is generalized to higher dimension lattices but most studies focus on the short-range correlated interactions. This paper consider different kinds of long-range correlated generalizations of the original SYK model and discuss their chaotic and transport behavior. The results are novel and interesting although in fact I have some concerns. I would like to suggest its publication after getting satisfactory explanations.

Requested changes

  1. Add some discussions related to the point 1, 2 and 3 in the "Weaknesses" section.
  2. I think it's better to add a derivation of Eq. (45) and (48) in an appendix along with some explanations.
  3. There are some typos in the draft. For example: below Eq. (20) there is an empty bracket (). The author may need to revise the draft.

  • validity: good
  • significance: high
  • originality: high
  • clarity: good
  • formatting: acceptable
  • grammar: excellent

Anonymous on 2018-05-21  [id 255]

(in reply to Report 1 on 2018-03-22)

This author appreciates the comments made by the referees who both pointed out the importance and originality of the present manuscript. In the revised version, some of their recommendations have been heeded to, albeit without affecting any of the results or conclusions.

Answering the comments by Referee 1:

  1. "When the correlation function of the interaction strength is given by (6), the model is reduced to an SYK model with NL fermions only if J~1/L^{32}. This also leads to the divergence when one combines (20) with (15), which can be canceled by choosing a correct scaling of J in terms of L. As a result the instability discussed below (20) may not be physical. (By the way, even if we do not add a scaling of L to J, I think the divergence can not be canceled by assuming the non-locality of Green's function.)"

    Author's reply:

    It is indeed true (and, in the original manuscript, was stated explicitly in the text following Eq.(30)) that in the SYK model defined on a dimensionless cluster of NL sites (where both N and L are treated as arbitrary integers) a proper normalization of the variance of the random amplitude J would have to be additionally scaled with 1/L^{q-1} in order to have a well-defined 'large NL' limit.

    However, in a setup that can be potentially relevant to the condensed matter applications the number of lattice sites L would be macroscopic (~10^24), whereas the number of 'orbitals' N would be, typically, of order one. And even though N can still be formally considered large for the purpose of utilizing the soluble large-N limit, treating L in the same manner would amount to studying a mesoscopic 'quantum dot' where - as opposed to a macroscopically extended periodic d-dimensional spatial lattice - translational invariance is not an all-important factor and transport of charge, energy, and/or momentum (or a lack thereof) can not even be defined without introducing proper boundary conditions mimicking the external 'leads'. By contrast, in a macroscopic lattice system the strength of the random correlations would have to be chosen independent of the system's size L, thus allowing for a smooth behavior at L\to\infty which is free of the finite-size effects.

    Some clarification to that effect was added to the text following Eqs.6,7.

    Thus, any divergence that stems from a slow spatial decay of the system's size-independent correlations would be a genuine 'infrared' one. In fact, one does not really have to strive to 'cancel the divergence' at all costs - but instead to identify the regimes where it would or would not occur in the first place. As was clearly stated in the original manuscript, such divergences associated with the 'Hartree' terms may or may not be present, depending on the chosen type of the random correlations (e.g., Eqs.8 and 9, both of which are equally L-independent). Moreover, the conditions under which such divergences would be absent could even serve as a criterion for generalizing the SYK model in a physically sound - rather than purely academic - way.

  2. "If we take Eq. (10) and use a diagrammatic approach starting from a non-interacting Green's function and summing up all diagrams, the Green's function will always be super-local. The author does not provide any evidence for the emergence of the non-locality (The overall factor of the Green's function is not given and the author does not compare the energy of different solutions). Or maybe the simplest way is just modifying the UV kinetic energy by adding some spatial derivatives to tune different z. Will this modify the effective action?"

    Author's reply:

    Any argument based on perturbative diagrammatic calculations is prone to failing in the case of intrinsically non-perturbative spontaneous symmetry breaking where one introduces new diagrammatic elements ('condensates') by hand and than explores the system's (in)stability in their presence. Examples of the latter include not only the best known case of a two-particle pairing function but also less familiar ones, including those of a single-particle nature, such as an inter-site ('off-diagonal' in real space) component of the single-particle propagator. One pertinent example is provided by a non-vanishing density of states of the zero-density Dirac fermions in the presence of potential disorder (P. A. Lee, Phys. Rev. Lett. 71, 1887 (1993)) which appears to vanish to any finite order of perturbation theory, thus signaling spontaneous breaking of chiral symmetry (above a finite threshold in d\geq 3 but without any in d\leq 2). Similar to the above, in the generalized SYK model a finite inter-site amplitude G_{ij} can be teased out by first introducing such a probe amplitude and then observing that the corresponding analogue of the 'gap equation' develops a non-trivial solution.

    Some clarification to that effect was added to the text following Eq.(24).

    As to the energies of the different (local vs non-local) solutions, in the original manuscript they were indeed evaluated and found to be equal in the mean field approximation (see Eq.30).

  3. "The Eq. (26) is derived by considering the kinetic term, but the Green's function is solved after neglecting the kinetic term. The logic here may need further explanation."

    Author's reply:

    As was stated in the original manuscript, searching for the propagator in the general scaling form (25) leaves the dynamical index z undetermined, and so Eq.(26) should be viewed as its limiting value for which Eq.(17) can still be satisfied, albeit only marginally. In this case neither the anomalous self-energy, nor the bare kinetic term dominate, while keeping both can potentially alter the numerical prefactor in (25) by a coefficient of order one (which is why an accurate calculation of this prefactor requires a numerical solution).

    Some further explanation to that effect has been added to the text following Eq.(26).

  4. "Some results (for example Eq. (45) and (48)) are given without derivation, which makes the paper less easy to understand. Taking (45) as a example, does the result means: the coefficient of the δ-term given by the inner product of wave function and the 1/J terms indicates the shift of eigenvalue of K by the UV kinetic term is proportional to |ω| (as the original SYK model)? Is there a simple scaling argument for the form of them?"

    Author's reply:

    This manuscript was not intended to cover the important aspects of the generalized SYK models all at once. Indeed, it took several dozens of multi-authored papers to exhaustively investigate even the (obviously, simpler) original SYK model (see References). In that regard, Eqs.(45,48) presenting the effective action for fluctuations should be viewed as largely schematic. In particular, obtaining accurate values of the coefficients in the $\delta$-term would be rather difficult, thus making it hard to provide a systematic and fully satisfactory technical derivation of such expressions.

    However, the origin of the novel terms is rather straightforward and, in particular, the term $|\omega|\delta$ appearing in Eq.45 should indeed be viewed as the shift of the unity eigenvalue of the ladder kernel (36) which results from breaking the d+1-dimensional reparametrization invariance by a temporal ($\alpha\neq 0$) and/or spatial ($\beta\neq 0$) dependence of the disorder correlations. In contrast to the bare kinetic term, though, such symmetry breaking occurs at arbitrarily large J and, therefore, the corresponding term lacks the extra $\omega/J$ factor, as compared to the case of the original SYK.

    An additional clarification to that effect was added to the text following Eq.(45).

  5. "There are some typos in the draft. For example: below Eq. (20) there is an empty bracket ()".

    The missing reference to Eq.19 was added to the text.

"Requested changes:

  1. Add some discussions related to the point 1, 2 and 3 in the "Weaknesses" section.

  2. I think it's better to add a derivation of Eq. (45) and (48) in an appendix along with some explanations.

  3. There are some typos in the draft. For example: below Eq. (20) there is an empty bracket (). The author may need to revise the draft."

The above points have all been addressed.

---

## Round 3 · Referee Report · Anonymous (Referee 2) · 2018-5-16

Strengths

1- The manuscript provides an interesting generalization of the SYK model to a class of non-local models in any spatial dimension 2- Thorough analysis of a few examples of the interactions correlations. The known limits (SYK model, q=2 non-interacting fermions, other generalizations of the SYK model) are carefully obtained

Weaknesses

1- There is missing scaling in (6) J -> J/L^(q/2-1/2) (the other referee took q=4), which can change the scaling in the remainder of the paper. 2- Stemming from this problem, it is not clear why is J taken L-independent constant for the other models considered in the manuscript 3- There is no discussion of local, but not ultra-local model. 4- The author generalizes the conclusions obtained for a concrete model to all SYK generalizations too easily 5- Scaling dimension discussion is very important, but I do not think that the models should be discarded solely on the basis of the interaction term being irrelevant. More relevant terms can be prohibited by symmetry

Report

I think the understanding of the different generalizations of the SYK model shows us how universal the connections of it to AdS/CFT, black hole physics, quantum chaos, etc. are. The results of the present manuscript are thus interesting enough to warrant publication after the concerns are addressed.

Requested changes

1- Fix the scaling of J in (6) and address the scaling of J's in (7-9) 2- Add discussion of the local (exponentially decaying) correlations. 3- Remove or reformulate the discussions of 'sickening' of the SYK model, in particular discuss the limitations of the results obtained for a few solvable models to the general class of the models defined by (5)

  • validity: high
  • significance: high
  • originality: top
  • clarity: good
  • formatting: perfect
  • grammar: good

Anonymous on 2018-05-21  [id 256]

(in reply to Report 2 on 2018-05-16)

Answering the comments by Referee 2:

  1. "There is missing scaling in (6) J -> J/L^(q/2-1/2) (the other referee took q=4), which can change the scaling in the remainder of the paper."

    Author's reply:

    See our reply to Referee 1's comment 1.

  2. "Stemming from this problem, it is not clear why is J taken L-independent constant for the other models considered in the manuscript"

    Author's reply:

    See our replies to Referee 1's comments 1 and 2.

  3. "There is no discussion of local, but not ultra-local model."

    Author's reply:

    Once again, the manuscript was not meant to provide a comprehensive coverage of all the potentially relevant aspects of the various generalized SYK models. Nevertheless, it did include a discussion of the different types of behavior which, depending on the value of \beta, would give rise to either the conventional diffusive dynamics (akin to that in the short-range-correlated 'SYK-lattice' model of Refs.13,42) or, else, a novel 'sub-diffusive' behavior.

    Some clarification to that effect was added to the text following Eq.(47).

  4. "The author generalizes the conclusions obtained for a concrete model to all SYK generalizations too easily."

    Author's reply:

    Although the manuscript's conclusions pertain to the specific family of disorder correlations described by Eqs.(8) and (20) or (31), this class of functions is sufficiently broad to suggest that even more general long-range correlations decaying slower than a certain critical power-law (evaluated above as $(d+2)/2$) could give rise to the behavior that is markedly different from that obtained in the generic short-range correlated 'SYK-lattice' models, including those with the 'same and nearest neighbor only' [13,42] as well as exponentially decaying random couplings.

  5. "Scaling dimension discussion is very important, but I do not think that the models should be discarded solely on the basis of the interaction term being irrelevant. More relevant terms can be prohibited by symmetry."

    Author's reply:

    It was not on this author's agenda to discard any physically sound models. In fact, somewhat ironically, so far he would seem to be in the absolute minority, advocating the need to venture off the well-beaten path of the ultra-local SYK physics and abandoning its comfort zone for the sake of discovering and classifying the new types of non-Fermi liquid behavior instead of dwelling 'ad nauseam' on the former.

    As regards the specific long-ranged generalizations of the SYK model discussed in the manuscript, they would not be apriori prohibited by any known symmetry. Of course, any definitive answers can only be obtained by investigating those generalized models more thoroughly - which, at the very least, requires them first to be deemed legitimate and worth of studying, regardless of the potential implications (which might indeed be somewhat disturbing, if contrasted against the orthodox SYK model).

"Requested changes:

1- Fix the scaling of J in (6) and address the scaling of J's in (7-9)

2- Add discussion of the local (exponentially decaying) correlations.

3- Remove or reformulate the discussions of 'sickening' of the SYK model, in particular discuss the limitations of the results obtained for a few solvable models to the general class of the models defined by (5)."

This author chooses not to follow the last referee's recommendation as, in his view, the use of the original title is justified by the logic of presentation and is not merely stylistic. In defense of such a decision one could also invoke an obviously independent (albeit dated by an almost one year after the original release of the present manuscript in its preprint form arXiv:1705.03956) use of the term 'sickening' (with very similar allusions and connotations) in the recent (May 7, 2018) talk of the founder of the SYK research - see http://qpt.physics.harvard.edu/talks/jqi18.pdf, slides 68-72, "Infecting a Fermi liquid and making it SYK".

---

## Round 4 · Referee Report · Anonymous · 2018-6-1

Report

In this revised version, the author properly answers most of my previous questions. The draft contains a thorough study on different generalizations of SYK models and is of high quality. Now I suggest the publication of this draft.

Nevertheless, I still have difficulty understanding the equation 45 and 48 in this revised draft although the author have explained a lot in the reply. For example, in the Lorentz invariant case (48), why the reparametrization field appears as (\partial_\mu \epsilon^\mu)^2 instead of \epsilon_\mu \partial^2 \epsilon^\mu? Why we have a factor k_\mu^{D/2} here? Similar questions can be asked for Eq. 45. I think it may be helpful to explain this in more details or add some references.

  • validity: top
  • significance: top
  • originality: top
  • clarity: high
  • formatting: perfect
  • grammar: perfect

Anonymous on 2018-06-21  [id 277]

(in reply to Report 1 on 2018-06-01)

Answering the comments by Referee 1:

"Nevertheless, I still have difficulty understanding the equation 45 and 48 in this revised draft although the author have explained a lot in the reply. For example, in the Lorentz invariant case (48), why the reparametrization field appears as (\partial_\mu \epsilon^\mu)^2 instead of \epsilon_\mu \partial^2 \epsilon^\mu? Why we have a factor k_\mu^{D/2} here? Similar questions can be asked for Eq. 45. I think it may be helpful to explain this in more details or add some references."

Author's reply:

It is indeed true that the corresponding quadratic forms might be more general than the ones presented in the schematic effective actions (45,48). In the revised version, we replaced those with the most general expressions that are quadratic in frequency and momentum (Lorentz invariance does impose some restrictions, though) and provided the necessary comments, following the above equations.

---

## Round 4 · Referee Report · Anonymous · 2018-6-13

Report

In the revised version the author answers my questions and implements changes except the last one suggested. I should have been more precise in the request. To me the pre-last paragraph of the manuscript, "Our findings suggest, however..." sounds as an unnecessary generalization and should be made into a weaker statement. One can change "could not serve" -> "unlikely to serve" for example. Outside of this issue I thank the author for the detailed answer and recommend the manuscript for publication

  • validity: -
  • significance: -
  • originality: -
  • clarity: -
  • formatting: -
  • grammar: -

Anonymous on 2018-06-21  [id 276]

(in reply to Report 2 on 2018-06-13)

Answering the comments by Referee 2:

"In the revised version the author answers my questions and implements changes except the last one suggested. I should have been more precise in the request. To me the pre-last paragraph of the manuscript, "Our findings suggest, however..." sounds as an unnecessary generalization and should be made into a weaker statement. One can change "could not serve" -> "unlikely to serve" for example. Outside of this issue I thank the author for the detailed answer and recommend the manuscript for publication."

As per the referee's advice, we somewhat softened the language of our final conclusions.

---

## Round 5 · Author Response

Dear Editors,

Following your advice, in the new revised version of the
manuscript we addressed the remaining concerns expressed by the referees during the second round of refereeing. Such modifications have neither affected any of our results, nor resulted in any changes to our conclusions. Below, we provided our answers to the referees' comments.
We do hope that the final version of the manuscript would now be publishable without a further delay.
Thank you for your consideration.

Sincerely yours,
Dmitri Khveshchenko

---

## Round 5 · List of Changes

Answering the comments by Referee 1:

"Nevertheless, I still have difficulty understanding the equation 45 and 48 in this revised draft although the author have explained a lot in the reply. For example, in the Lorentz invariant case (48), why the reparametrization field appears as (partial_mu epsilon^mu)^2 instead of epsilon_mu partial^2 epsilon^mu? Why we have a factor k_mu^{D/2} here? Similar questions can be asked for Eq. 45. I think it may be helpful to explain this in more details or add some references."

Author's reply:

It is indeed true that the corresponding quadratic forms might be more general than the ones presented in the schematic effective actions (45,48). In the revised version, we replaced those with the most general expressions that are quadratic in frequency and momentum (Lorentz invariance does impose some restrictions, though) and provided the necessary comments, following the above equations.

Answering the comments by Referee 2:

"In the revised version the author answers my questions and implements changes except the last one suggested. I should have been more precise in the request. To me the pre-last paragraph of the manuscript, "Our findings suggest, however..." sounds as an unnecessary generalization and should be made into a weaker statement. One can change "could not serve" -> "unlikely to serve" for example. Outside of this issue I thank the author for the detailed answer and recommend the manuscript for publication."

As per the referee's advice, we somewhat softened the language of our final conclusions.

Docutils System Messages

---

## Editorial Decision

published